# A national extent map of cropland and grassland for Switzerland based on Sentinel-2 data

Robert Pazúr[1], Nica Huber[1], Dominique Weber[1], Christian Ginzler[1], Bronwyn Price[1]

[1]Swiss Federal Institute for Forest, Snow and Landscape Research WSL, Zürcherstrasse 111, CH-8903 Birmensdorf, Switzerland

*Correspondence to*: Robert Pazúr (robert.pazur@wsl.ch)

**Abstract.** Agricultural landscapes support multiple functions and are of great importance for biodiversity. Heterogeneous agricultural mosaics of cropland and grassland commonly result from variable land use practices and ecosystem service demands. Switzerland's agricultural land use is considerably spatially heterogeneous due to strong variability in conditions, especially topography and climate, thus presenting challenges to automated agricultural mapping. Nation-wide knowledge of the location of cropland and grassland is necessary for effective conservation and land use planning. We mapped the distribution of cropland and permanent grassland across Switzerland. We used several indices largely derived from Sentinel-2 satellite imagery captured over multiple growing seasons, and parcel-based training data derived from landholder reporting. The mapping was conducted within © Google Earth Engine using a random forest classifier. The resulting map has high accuracy in lowlands as well as in mountainous areas. The map will act as a base agricultural land cover dataset for researchers and practitioners working in agricultural areas of Switzerland and interested in land cover and landscape structure. The map as well as the training data and calculation algorithms (using © Google Earth Engine) are freely available for download on the Envidat platform, doi: 10.16904/envidat.205 (Pazúr et al., 2021).

## 1 Introduction

Cropland and grassland cover 34% of the earth's terrestrial surface and represent the second most widespread land cover (LC) class, after forests (36%) (Buchhorn et al., 2020). Cropland and grassland provide multiple services for humans and nature, for example, food and fodder provisioning, habitat for various species, or cultural heritage (Bengtsson et al., 2019). The provisioning of these services varies substantially across the globe and is influenced by climate, cultural factors and the spatial and temporal configuration of landscape types. Areas with multiple use demands require a high degree of landscape multifunctionality. In multifunctional landscapes, the spatial allocation of cropland and grassland is strongly influenced by management policies supporting production and protection services in different forms and structures (e.g., grassland subsidies for management practices, usually related to mowing, grazing and fertilizing regimes). Sustainable management strategies may help to maintain vulnerable ecosystems in agricultural areas, increase biodiversity, or minimize the risks associated with inappropriate management (e.g., soil erosion or degradation of grassland ecosystems) (Wezel et al., 2014). Therefore,

knowledge of the spatial mosaic of cropland and grassland is extremely important. Such maps also determine the design of ecological networks and evaluation of current and future management strategies.

Accurate maps of agricultural areas at high spatial resolution and available at national scales have previously been very rare. Only recently has it been possible that the demands for such maps of agricultural areas could be met through the undertaking of several national or international projects, thanks to the increasing availability of open access remote sensing data. For

example, mapping of grasslands and their management has been conducted at the national extent for Germany (Griffiths et al., 2019). A grassland map is also part of the Copernicus Land Monitoring Service High Resolution Layer group for Europe produced by European Environmental Agency (Copernicus Land Monitoring Service, 2020). In comparison to European scale products, nation-wide products can benefit from better parametrization of classification model(s) to account for the sub-continental variation within agricultural areas (e.g., in field size and mosaics) which occurs within Europe due to different

management practices or climatic conditions.

We mapped cropland and permanent grassland across the whole of Switzerland. Switzerland covers 41 285 km2 and is well suited to the development and testing of methods for mapping agricultural land cover due to the availability of extensive ground-truth data. Despite being a small country, Switzerland is very heterogeneous with respect to climate and terrain conditions which have a strong influence over the ecosystems services provided by agricultural land (Fig. 1), as well as in

socio-political terms with variation across the 26 cantons. Switzerland is divided into 6 biogeographical regions (Fig. 1) defined following statistical analysis of the distribution of flora and fauna species and adapted to boundaries of communes (Gonseth et al., 2001; Wohlgemuth, 1996). These regions differ significantly in their climate and topography. The demand for ecosystem services provided by agricultural areas is relatively high compared to other European countries since the import of agricultural goods is relatively limited (compared to EU member states) and protection measures are applied across agricultural areas.

Protection measures include designation of strict protection areas where fertilization and irrigation are banned, and measures that control the timing and number of management activities such as mowing, grazing and fertilization of grasslands (Boch et al., 2019). In addition, direct payments schemes are also used to manage arable land requiring implementation of crop rotation and managed fertilizer regimes, restrictions on pesticides use and compulsory implementation of buffer zones and ecological compensation areas (FOAG, 2015). As a result, areas which are topographically suitable for arable agriculture (which are

relatively limited) are likely subject to intensive farming.

We used Sentinel-2 imagery from the growing seasons of 2017-2019 to map cropland, permanent grassland and shrubland at 10-meter resolution. Although we tested a model using imagery from a single year (2019), we found that this approach resulted in lower accuracies than multi-year data. Since our aim was to differentiate permanent grassland from cropland, temporary/annual grassland was considered part of the crop rotation and as such multiyear imagery was more fit to our purpose.

We trained and parameterized a random forest classifier within the © Google Earth Engine (GEE), using multiple Sentinel-2 indices on selected agricultural parcels from all over Switzerland, for which information was available on the crop and grassland types cultivated in a specific year. Using a cross-validation (split-sample) approach we found that the resulting map reached an overall accuracy of 84-95% in the Jura and Plateau region and 90-95% in Alps. The misclassified pixels were

mostly grasslands falsely assigned to cropland areas and mostly related to annual grasslands, which were considered to be
cropland within the training data. The resulting map of cropland (including annual grassland) and permanent grassland
provides a spatially explicit, area-wide alternative to the only existing national dataset, the Swiss Land Use Statistics, which
is a point-based statistical dataset on a 100m grid (Bundesamt für Statistik, 2020). The application combines two models, one
stratified for the Alps and one for the areas outside of the Alps, and as such the trained model is also applicable within different
biomes. There is potential for the trained model and classification algorithm to be transferable to map cropland and grassland
in different growing seasons or in different countries assuming similar agricultural management practices and climatic
conditions.

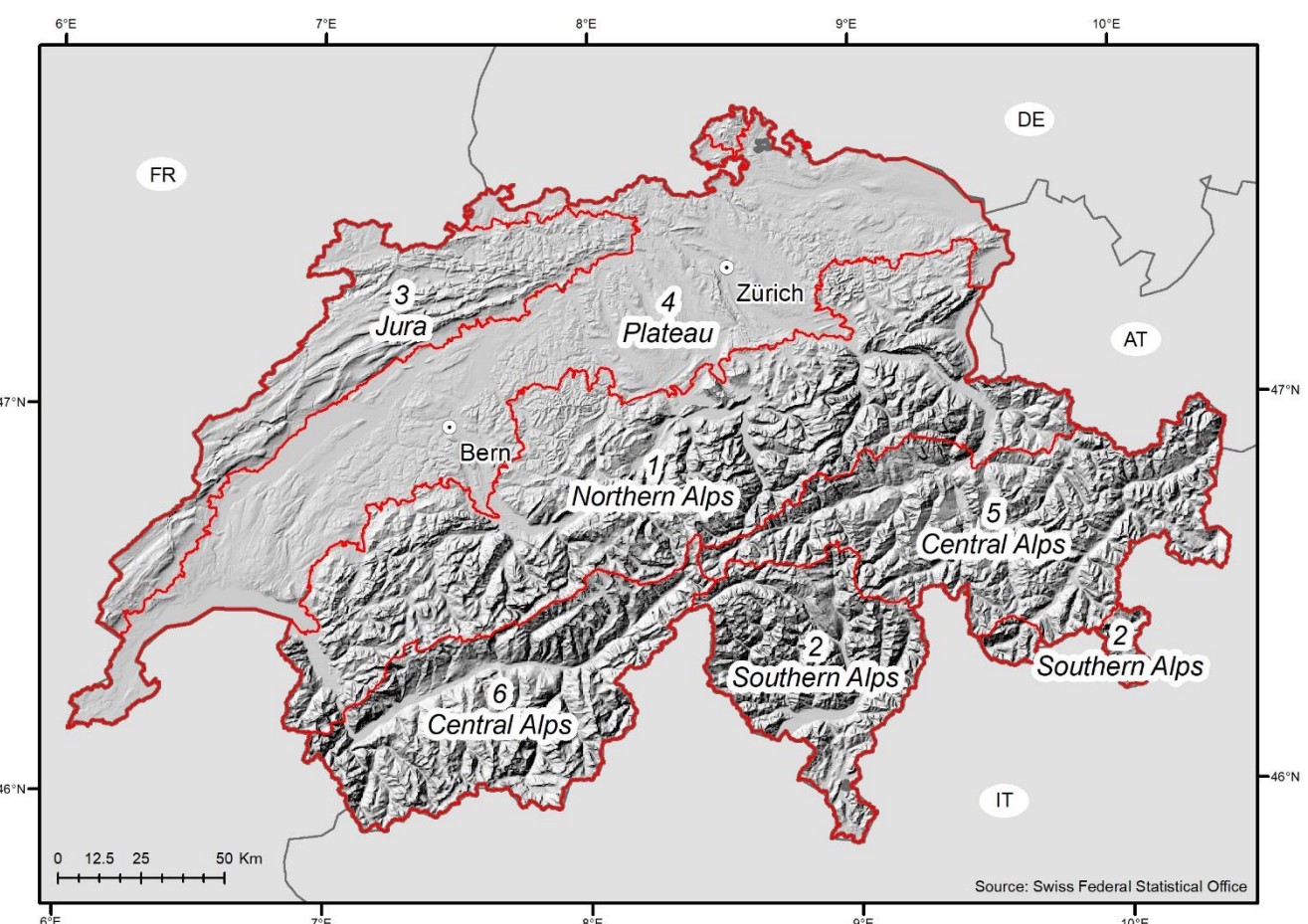

**Figure 1: Biogeographic regions and shaded relief of Switzerland**

## 2 Data and Methods

### 2.1 Satellite imagery

To identify cropland, grassland and shrubland, we used spectral information retrieved from all Sentinel-2 pixels available for Switzerland between April-November of 2017-2019 with initial cloud coverage lower than 80% (as reported in the Sentinel-2 metadata). We used the Sentinel-2 imagery preprocessed to surface reflectance (Level 2A) within GEE (GEE repository ee.ImageCollection("COPERNICUS/S2_SR")) and applied a cloud masking procedure (GEE repository ee.ImageCollection("COPERNICUS/S2_CLOUD_PROBABILITY")). This procedure uses deep learning semantic segmentation (Garcia-Garcia et al., 2017) to set the cloud occurrence probabilities within Sentinel-2 scenes (Zupanc, 2017) and has been documented to outperform other cloud masking procedures. Additional scripts were used to mask shadows and snow coverage (see the linked GEE codes for more information). After preprocessing, the number of images available within the time period varied substantially over Switzerland due to cloudiness, especially in the Alps, and the acquisition pathways of the Sentinel-2 satellite sensors (Fig. 2).

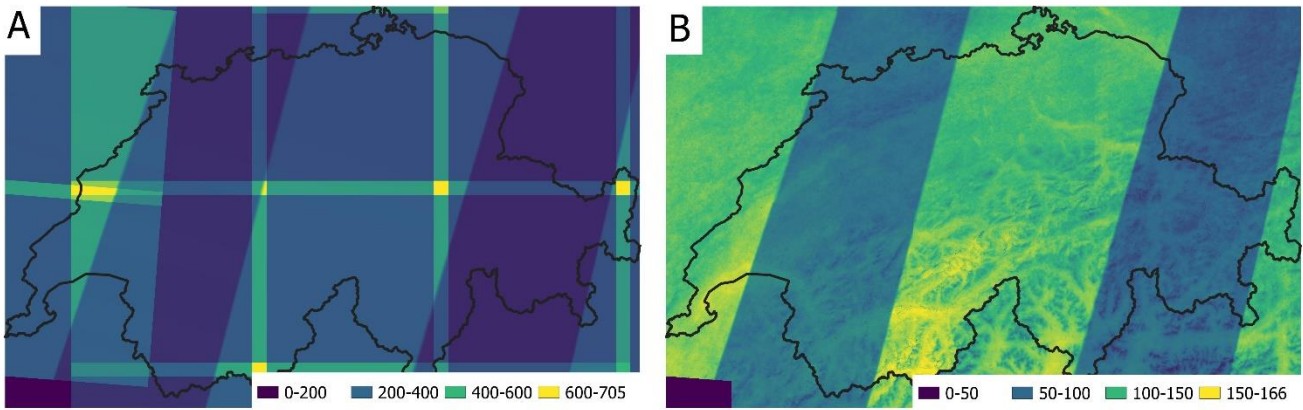

**Figure 2: The number of available Sentinel-2 scenes (A) prior to the cloud masking including tile edge effect and (B) after applying the cloud mask and masking of the scene edges.**

### 2.2 Training data

To identify the spectral parameters of cropland and grassland, we used parcel-level training data indicating the occurrence of different crop types and permanent grassland derived by cantonal (state) authorities based on the reporting of farmers in the years 2017-2019. We've considered parcels with reported crop usage (e.g., wheat, spelt, corn, potatoes, sugar beets) within a reported year to be cropland. Due to the practise of crop rotation which is frequently applied on Swiss agricultural parcels, as well as different crop type within an individual field, it is likely that the spectral characteristics of a given single parcel changed within the 3-year study period. However, the training data for each parcel is for one time point only within the study period and therefore such temporal variations were not considered within the training data. Cropland is considered to be an area covered by crops within a given year for particular training data while grasslands are considered to be areas explicitly reported

as permanent grasslands. Permanent grasslands are likely to have similar phenological trajectories over the multiple growing seasons considered within the study period. Parcels classified as annual grasslands within the reported statistics were considered to be cropland because we considered annual grassland to be a part of the crop rotation management.

The parcel-level training polygons were selected manually from different areas across Switzerland. The overall requirement for selection of a polygon was its visual homogeneity (e.g., no trees within the cropland or grassland fields). We also considered the variability of croplands and grasslands by including parcels that reported different land use within those two classes (e.g., different cropland types). To avoid selecting pixels of mixed land use, which are likely to occur in heterogeneous landscapes, especially at parcel edges, we shrunk the training polygons by a 10m "inside" buffer distance (buffer size defined according to the resolution of Sentinel-2 bands). In total, 1378 polygons covered by more than 400'000 10m pixels were selected to train the classification model.

In addition to the cropland and grassland data, we provided the RF model with training data on shrubland, forest, wetland, water surfaces. This training data allowed us to better define the ranges of reflectance of different LC in the training data. In the accuracy assessment we only include the shrubland as it's a widespread LC in Swiss mountainous areas and, due to its vegetation cover with minimal seasonal variation in reflectance, is likely to be considered grassland by our predictive models. Other LC classes were further masked out using different maps and were not an object of accuracy testing. Training data on different LC classes were generated from 70, 25, 13 polygons with an average area of 11ha, 30ha, 1177ha for shrubland, wetland and water surfaces, respectively. These polygons were digitized manually from the Sentinel-2 imagery guided by the Swiss Land Use Statistics.

### 2.3 Classification indices

We extracted multiple indices for every Sentinel-2 pixel overlapping the training polygons. The indices used were chosen based on previous mapping of Swiss agricultural areas (Kolecka et al., 2018; Pazúr et al., 2021) and followed general assumptions about the differences between the phenology of cropland and grassland over a year (e.g., low growth on cropland in autumn), low variability of permanent grassland over time. The selected indices characterize phenology either over the whole growing season or over a particular part of the growing season, or relate to summary statistics of the spectral bands of Sentinel-2 (Table 1). Similar indices have been found useful for land cover classification over large areas (Pflugmacher et al., 2019) and mapping phenological responses in grassland area or cropland types (Ghazaryan et al., 2018; Gómez Giménez et al., 2017). Furthermore, we included variables characterizing the terrain properties of each pixel, such as elevation, slope and terrain orientation retrieved from the 30m Shuttle Radar Topography Mission data (GEE repository ee.ImageCollection ("USGS/SRTMGL1_003") (Farr et al., 2007).

**Table 1 Indices used in the classification model as calculated from the growing seasons of 3 years of Sentinel-2 satellite data**

| Index | Explanation |
| --- | --- |
| **Single band indices** | |
| BLUE, GREEN, RED, NIR, SWIR1, SWIR2 | median of the Sentinel-2 band values |
| **Indices characterizing the phenology over the growing season** | |
| ndvi, evi | median of NDVI, EVI phenological indices |
| ndvi_stdDev | standard deviation of the NDVI |
| ndvi_kurtosis | kurtosis of the NDVI curve |
| ndvi_skewness | skewness of the NDVI curve |
| **Indices characterizing phenology over a particular part of the growing season** | |
| ndvi_pc_05, ndvi_pc_25, ndvi_pc_85, ndvi_pc_95 | $5^{th}$, $25^{th}$, $85^{th}$ and $95^{th}$ percentile of the NDVI values NDVI recorded through all three-years of Sentinel-2 data, respectively |
| sSpring, sSummer, sAutumn | medians of NDVI within the 4, 5-8 and 9-11 month, respectively |
| ndviIntMean | mean of the NDVI values within the $50^{th}$ and $95^{th}$ percentile |
| ndviDiff | count(NDVI>0.3)-count(NDVI>0.7) |
| **Terrain related indices** | |
| elevation | elevation |
| slope | slope of the terrain in degrees |
| aspect | orientation of the terrain |

## 2.4 Classification

To account for different environmental conditions (i.e. climate and terrain) within Switzerland, the classification models were calculated separately for two strata defined by biogeographic regions. The biogeographical regions of the Jura Mountains and the Swiss Plateau together form one stratum (regions 3 and 4 on Fig. 1), while the Alpine regions form a second stratum (regions 1, 2, 5 and 6 on Fig. 1). While we tested several approaches to defining the strata, including a single national model and different combinations of the biogeographical regions, we found that the two strata chosen gave the best results and most

straight forward approach to taking into account the climatic and topographical differences between the more lowland areas and the mountainous areas. In particular the whole of Switzerland approach resulted in biases for crop detection towards the Plateau and higher inaccuracies in mountainous areas, particularly in mixed shrubland grassland areas.

We used the random forest (RF) classifier implemented within the GEE platform (library ee.Classifier.smileRandomForest)

for the classification of cropland, grassland and shrubland. The default parameters of the GEE algorithm were used except for the number of trees (300 trees were used) and the number of variables per split (10 variables per split were used). These variables were set according to the accuracy cross-validation checks of different combinations of their values (Kuhn, 2008). We also produced a separate layer of non-vegetated areas. i.e., sealed surfaces, rocks, and water bodies for masking purposes. Assuming that those surfaces have little or no phenological response, we classified these areas using the 95th percentile of the

distribution of all NDVI values recorded over the mapping period. Using two different thresholds, we found the non-vegetated areas under 2000 m a.s.l. on those pixels where the 95th percentile of recorded NDVI value was lower than 0.7. In areas above 2000 m a.s.l., the non-vegetated areas occurred where the 95th percentile of recorded NDVI value was lower than 0.6 (Fig. 3).

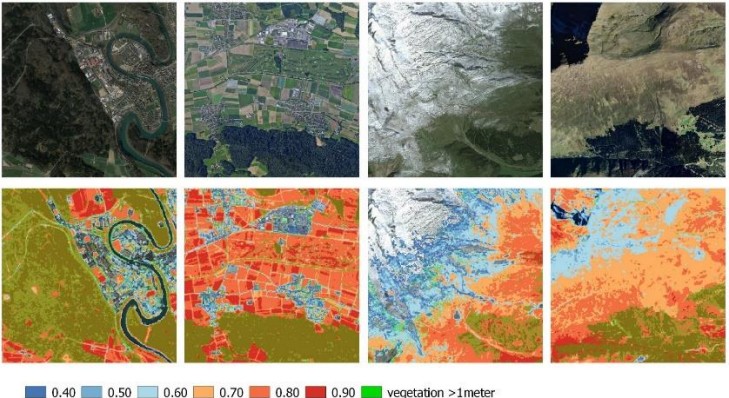

**Figure 3: Illustration of the performance of different layers considered to mask out sealed areas, water surfaces and rocks from the resulting map. The values represent different NDVI value thresholds observed at the 95th percentile. Green outlines show the areas of vegetation taller than 1 meter (Source of the aerial imagery: © Google Maps).**

## 2.5 Postprocessing

To increase the accuracy of the final map we masked out woody vegetation (forest, tree lines and single trees) from the model output with the vegetation height model (VHM) of Switzerland (Ginzler and Hobi, 2015). Within areas classified as cropland, woody vegetation was masked out using a vegetation height threshold of 3 meters. This was done to prevent the exclusion of crops that potentially reach 3 meters in height, especially towards the end of the growing season (the period when the imagery on which the VHM is based were captured). Within grasslands, a lower threshold of 1 meter was applied in order to mask out vineyards, shrubs and hedgerows. In addition, we converted all cropland areas smaller than 2000 m$^2$ to grassland as we considered this a minimum size for cropland parcels. This allowed us to eliminate the effects of misclassification of LC on mixed pixel areas, such as in settlement hinterland or at high elevations, where small patches of grassland may appear within areas of low vegetation cover e.g., in the transition zone between grassland and rocks. As it is likely that all the areas of forest were masked out using the VHM, we converted all the unmasked pixels of forest to shrubland as we found large uncertainties of distinguishing between those two classes within our model.

## 2.6 Accuracy assessment

Map accuracy was assessed against three separate testing datasets, the 'parcel-level+' testing dataset, derived from parcel level data and digitized polygons of other LC classes  derived a similar manner to that described in the 2.2 Training data section, and two independent data sets, the ALL-EMA ("Agricultural Species and Habitats' Monitoring Programme") (Riedel et al., 2018) testing dataset and the Swiss Land Use Statistics testing dataset, described further below. Since these datasets were produced independently and for differing purposes, each dataset has differences in extent, nomenclature and ground-truth information.

### 2.6 .1 Parcel-level testing dataset

To assess map accuracy using the parcel-level data, we trained separate random forest models (one for Jura and Plateau and one for Alps) using the R software random forest package (Liaw and Wiener, 2002) and similar training data and settings to the classification in GEE. In this case, the manually generated training dataset and digitized shrubland areas used to train the classification in GEE were split into training and testing datasets (80% of the polygons were used to train and 20% to test the model). Using a separate model allowed us to avoid redundancies associated with using the same data for training and testing (i.e. avoid to use the map that was produced from the full training dataset). To limit spatial clustering, we further limited the selected samples in each biogeographical strata by maintaining a minimum distance of 30 meters between selected samples.

The accuracy of the outputs was measured using the overall accuracy and the related sensitivity and specificity rates. In our case, sensitivity is defined as the proportion of correctly classified presences of grassland for the Jura and Plateau region, or of each class for the Alps region. In contrast, specificity is defined as the proportion of correctly classified absences of a

particular class. Class level accuracies were calculated since they do not inherit the bias that may be present in an overall accuracy measure (Foody, 2020).

**2.6.2 All-EMA testing dataset**

ALL-EMA, the 'Agricultural Species and Habitats' Monitoring Programme' (Riedel et al., 2018, 2019), is a Swiss monitoring programme designed to monitor biodiversity on agricultural land. Among other measurements, it records habitat and land use type on 170 1km$^2$ area squares through field visits, with an average of 19 10m$^2$ plots monitored on each of these squares (Riedel et al., 2018). Only data points from ALL-EMA –locations of cropland, grassland and shrubland habitat which overlap
the output map were included in the comparison.

**2.6.3 Swiss Land Use Statistics**

The Swiss Land Use Statistics dataset is a nation-wide point sample of land use and cover information on a 100m grid across Switzerland (Riedel et al., 2018, 2019). The standard nomenclature of the Swiss Land Use Statistics dataset comprises 72 unique categories combining land use and cover classes and each point of the dataset is interpreted visually from aerial
photography and additional datasets. This approach ensures relatively accurate and robust land use and cover information for a given point and date (generally defined by the capture date of the aerial imagery). The Swiss Land Use Statistics dataset was used to assess the spatial pattern of accuracy of our cropland and grassland map within blocks of 1km$^2$ in size (100 interpretation points per block). The 1km$^2$ blocks allowed us to identify the areas of agreement and mismatch between both datasets. The Swiss Land Use Statistics dataset is updated every 12 years, the current cycle 2013-2018 is still being processed
and not yet available for the full extent of Switzerland. Therefore, for the nation-wide assessment of cropland and grassland areas, we complemented the 2013/2018 dataset with data from the 2004/2009 survey areas where data from the current survey were not yet available (circa 17%). The following classes were considered to be grassland: natural grassland, farm pasture, mountain meadow, alpine, Jura and sheep pasture and unproductive grassland if covered by grass. The Swiss Land Use Statistics cropland class was used for crops.

**3 Results and Discussion**

**3.1 Classification indices**

The distribution of the values of the indices derived from Sentinel-2 and the elevation model within the generated samples indicated differences between cropland, grassland and shrubland (Fig. 4). Substantial differences between these LC classes were found for individual spectral bands (NIR, SWIR1, SWIR2), phenological indices (NDVI, EVI and its derivatives), and
215 terrain indices (elevation, slope).

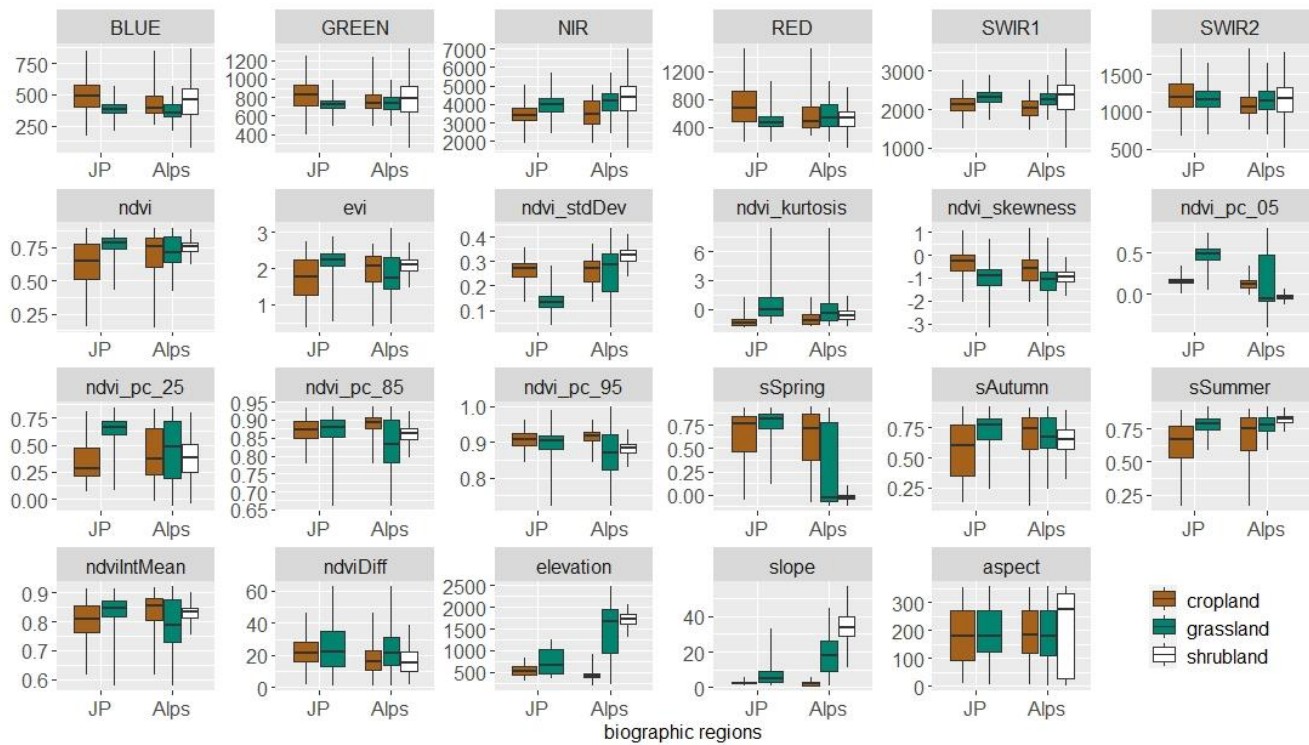

**Figure 4: Value ranges of indices within the training samples used to model the allocation of cropland and grassland stratified by biogeographic regions of (JP) Jura and Plateau and (Alps) Alps. For indices abbreviation details please refer to Tab. 1.**

Using those indices, we found better separability of cropland and grassland for the Jura and Plateau stratum compared to the Alps. Indices that characterize phenology over the growing season generally have higher values on grassland areas than cropland areas except for the standard deviation of NDVI, which is lower on grassland. For the Jura and Plateau stratum the standard deviation of NDVI was found as one of the most important indices in the classification model(Fig. 5).Grasslands within both stratas were also characterized by a negative skewness of the distribution of NDVI values. This relatively high proportion of higher NDVI values reflects the higher greenness of these areas. For shrubland, the values of these indices were within a very narrow span of the value distribution. This suggests the phenological behaviour of shrubland over the growing seasons is less variable than that of cropland or grassland. Good separability between cropland and grassland was also possible through the phenological indices ndvi_pc_05, ndvi_pc_25, which represent the lower percentiles of the distribution of NDVI recorded through all three-years of Sentinel-2 data. Especially for the Jura and Plateau stratum, both indices were much higher on grassland than on cropland and highly contribute to the accuracy of the resulting classification model. Within the samples generated for the Jura and Plateau stratum, using the single phenological indices, such as the ndvi_pc_05, may successfully separate cropland and grassland classes. By ensuring the robustness of such metrics e.g., by including multiple growing seasons, the model generated for the Jura and Plateau stratum may be applicable on flat and less complex terrains elsewhere.

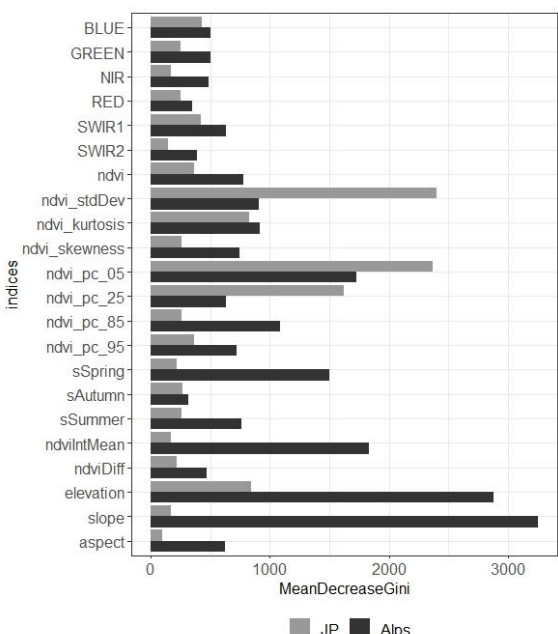

**Figure 5: Importance of different indices in the classification model within the Jura and Plateau stratum (JP) and the Alps (Alps) quantified using the mean value of gini-impurity loss function coefficient. More important indices in the model achieved higher values of the coefficient.**

However, we observed a wide range in spring seasonality (sSpring), especially on Alpine grasslands. This observation may be due to the low phenology of the alpine grassland at the beginning of the growing season or snow artefacts, which were missed from the snow mask applied in the classification especially during the snow melting period. Elevation also substantially aided the separation of grasslands from cropland and the classification accuracy, particularly within the Alps stratum. This observation can be considered relevant to mountainous areas, where cropland areas are generally only feasible at comparably lower elevations. Generally, the complex terrain conditions of the Alps, described by the terrain parameters, substantially influenced the separability of the cropland and grassland classes which might limit the transferability of the model outside of the Swiss Alps.

## 3.2 Accuracy assessment

Comparison between the derived map and the selected test datasets showed high accuracy of the modelling outputs (Fig. 6). The comparison of the output map with the test data derived from the selected parcel-level+ polygons resulted in the highest overall accuracies, 0.93 for the Jura and Plateau region and 0.92 for the Alps region. This accuracy assessment (parcel-level+ data) was also based on samples that were the most equally distributed across the different classes (prevalence ratios). The lowest accuracies were found for the comparison to the ALL-EMA dataset in the Alps regions for presence of cropland and shrubland, 0.44 and 0.31 (sensitivity ratio) and absence of grasslands, 0.38, (specificity ratio). The prevalence rates, however,

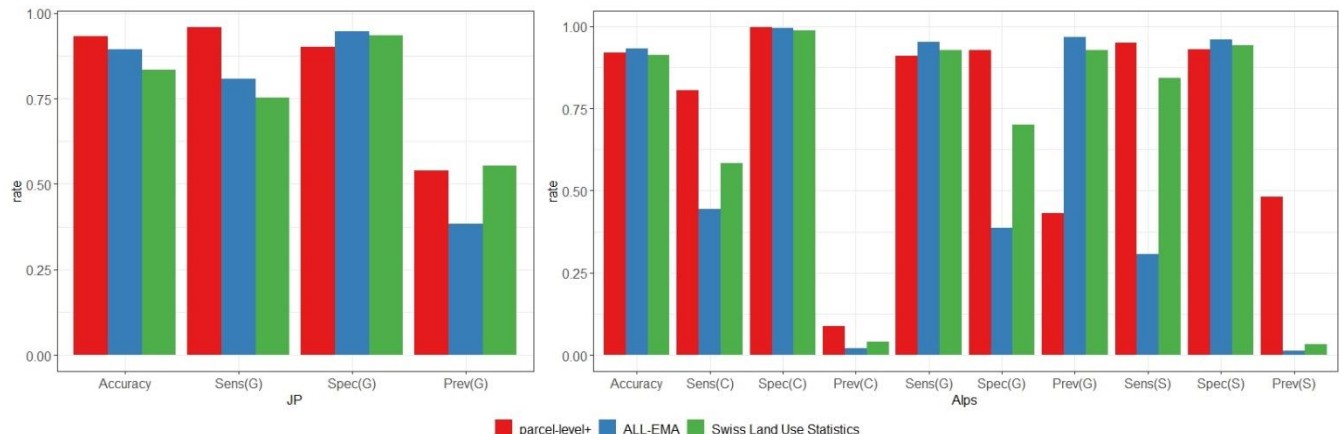

**Figure 6: Accuracy rates of the classification of (C) cropland (G) grassland on Jura and Plateau region (JP) and (C) cropland, (G) grassland and (G) shrubland on Alps (Alps). Shrubland was not considered in the Jura and Plateau region, the Sensitivity values (Sens(G)) reflect the correctly classified presence of grassland and the Specificity (Spec(G)) the absence of grassland, i.e. presence of cropland. The proportion of presence of grassland within the samples is indicated by Prevalence (Prev).**

**Table 2. Error matrix based on the ratios of true false observations**

| Map | Reference | | | | | |
| --- | --- | --- | --- | --- | --- | --- |
| | Other LC classes | Cropland | Grassland | Total [%] | User's Accuracy | n samples |
| Other LC classes | 58.9 | 0.3 | 4.1 | 63.3 | 93.1 | 2615k |
| Cropland | 0.3 | 6.8 | 0.6 | 7.7 | 87.8 | 318k |
| Grassland | 7.2 | 2.3 | 19.4 | 28.9 | 67.1 | 1194k |
| Total [%] | 66.5 | 9.4 | 24.1 | 100 | | |
| Producer's Accuracy | 88.7 | 72.1 | 80.5 | | | |
| n samples | 274k | 389k | 995k | | | |
| Overall Accuracy | 85.1 | | | | | |

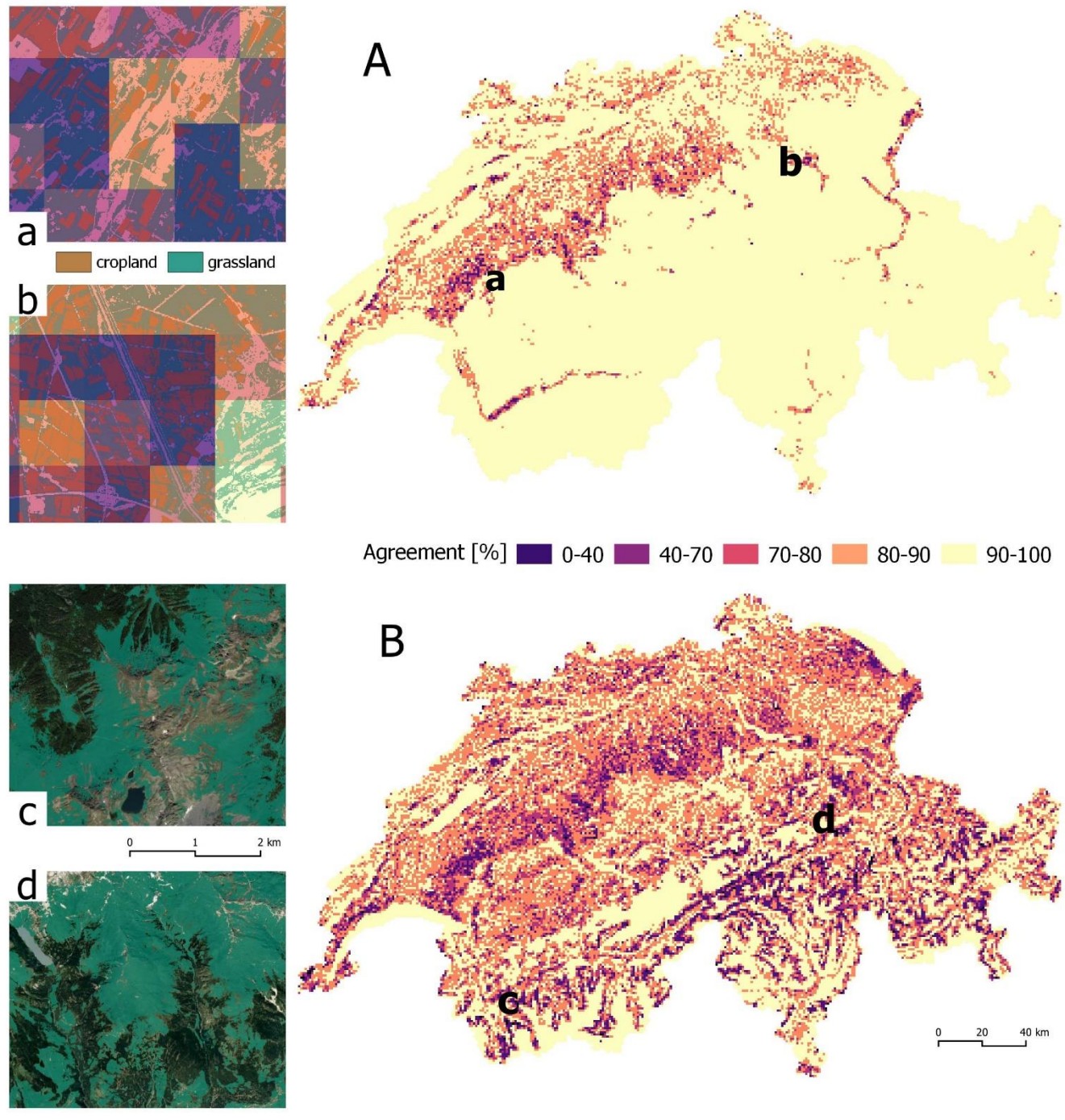

Fig. 7: Agreement [%] between (A) the cropland map and the reference classification (Swiss Land Use Statistics) and between (B) the grassland map and the reference classification (Swiss Land Use Statistics) per 1km$^2$ block (100 interpretation points per block). The rectangular zooms show the distribution of cropland and grassland (a,b) and grassland (c,d) in selected areas with less precise classification outcomes.

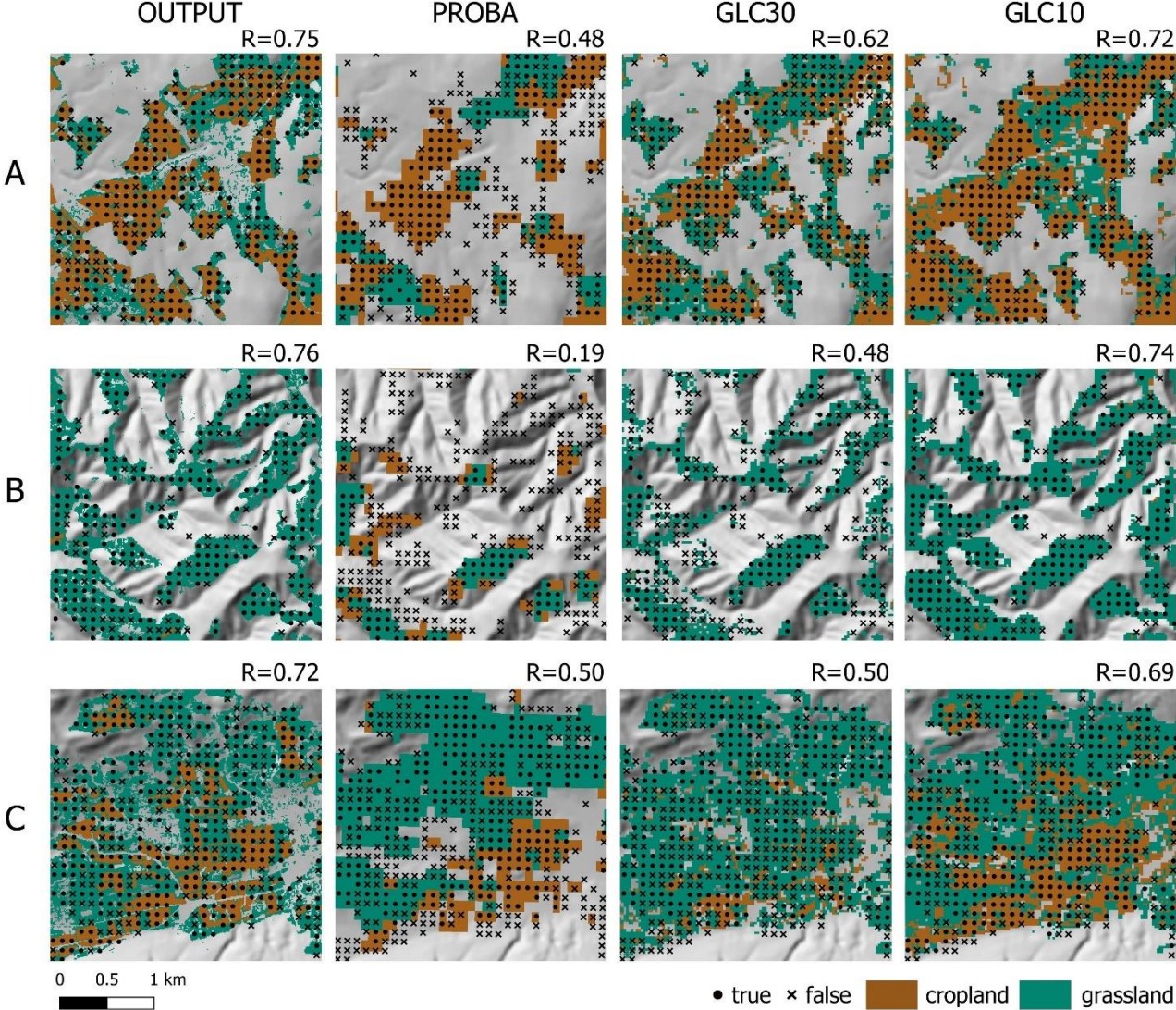

**Fig. 8: Illustration of the agreements between our cropland and grassland mapping output (OUTPUT), various land cover maps with global coverage (PROBA, GLC30, GLC10) and the reference classification (Swiss Land use Statistics, true and false symbols) on selected areas representing lowland (A), mountainous (B) and valley-like (C) terrain conditions. The R statistics defines the producer accuracy.**

demonstrate that these results relate to a limited number of samples. Due to low occurrences and a large number of samples, the prediction of absence of cropland in the Alps was of highest accuracy (0.99, specificity ratio) compared to the ALL-EMA testing data. The accuracy assessment of the shrubland mapping resulted in similarly high accuracy rates for both the parcel-level+ data and the Swiss Land Use Statistics. These similarities are to be expected since the parcel-level+ training data for

shrubs were selected from a subset of the Swiss Land Use Statistics. These accuracy assessments only consider areas of overlap between the output map and cropland, grassland and shrubland as defined by the given testing dataset. Potential bias of the output map outside of this extent, e.g., occurrence of cropland, grasslands and shrublands in areas of woody vegetation, lakes or non-vegetated areas, is not considered.

Furthermore, we assessed the nation-wide accuracy of cropland and grassland mapping using all Swiss Land Use Statistics interpretation points (n=4'128'498). This accuracy assessment resulted in high overall agreement with our classification at 85% with user's accuracy of 88% and 67% and producer's accuracy of 72% and 81% for cropland and grassland, respectively(Table 2). The lower accuracy may be due to nomenclature differences, which mean that the definitions of cropland and grassland are not fully aligned (e.g., agricultural crop rotation between crop and grassland on the same parcel). Moreover, the Swiss Land Use Statistics are determined from aerial imagery captured over an extensive time span which, depending on location, does not match with the period of our mapping data and the spatial resolution of the output map (10m). By summarizing the spatial distribution pattern of agreement/mismatch within 1km$^2$ blocks (Fig. 7), a higher degree of mismatch was found in mountainous and sparsely vegetated areas, and in wine growing areas. By comparing the cropland and grassland with the Swiss Land Use Statistics our mapping outcome perfomed well also in comparison with different land cover mapping products with global coverage (Fig. 8). Specifically, in case of accuracies of cropland and grassland mapping out output outperformed the Copernicus Global Land Cover 100m (PROBA, Buchhorn et al., 2021), Global Land Cover Map in 30m resolution (GLC30, Zhang et al., 2021) and Global Land Cover Map in 10m resolution (GLC10, Gong et al., 2019).

## 4 Data Availability

The output map (stored in the GeoTIFF eight-bit unsigned integer file format) identifies areas of cropland (value 1), permanent grassland (value 2) and shrubland (value 3) at 10-meter resolution over the entire area of Switzerland. Areas of shrubland, tall vegetation (forest and trees), sealed surfaces, water and rocks are masked out. Furthermore, patches of cropland smaller than 2000 m2 have been converted to grassland because they are assumed to be a result of misclassification. To apply these type of conversions that rely on the definition of raster patch sizes, we recommend the r.li toolset implemented in GrassGIS and QGIS software (Neteler et al., 2012; QGIS, 2020), or the RegionGroup tool in ArcGIS software (ESRI, 2016).

Since the unmasked map of cropland, grassland and shrubland might be helpful for certain ecological applications, we also published a version of the map with unmasked shrubland as well as the masks of non-vegetated areas (see the classification section of this paper). The resulting maps and codes are available on the EnviDat portal (Pazúr et al., 2021) under the following link: doi:10.16904/envidat.205.

The GEE code also links the datasets used to train and mask the output map.

## Author contributions

R.P.: Conceptualization, methodology, validation, software, visualizations, writing - original draft preparation. N.H.: Conceptualization, methodology, writing - review and editing. D.W.: Methodology, validation, writing - review and editing. C.G.: Conceptualization, writing - review and editing. B.P.: Conceptualization, methodology, writing - review and editing.

## Competing interests

The authors declare no competing interests.

## Acknowledgements

This dataset was developed within the project 'Erstellung einer Lebensraumkarte Schweiz 2019-2021' conducted at the Swiss Federal Research Institute WSL and funded by the Swiss Federal Office for the Environment.

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
