# Peer review of "A national extent map of cropland and grassland for Switzerland based on Sentinel-2 data"

_Earth System Science Data, 2021_

## Author Response (AR1)

| Reviewer #1 comments | Author's reply |
|---|---|
| About the data set used for classification. (1) Sentinel-2 data of three years (2017~2019) were used for mapping cropland and grassland, so that are there landcover changes in the selected years? As far as I am concerned, we need an annual cropland map for agricultural applications, I am puzzled whether this cropland/grassland map is applicable. | The purpose of this study was to differentiate permanent grassland from the areas of different crops. However, due to the crop rotation cycle, some cropland areas may be covered by temporary grassland for a single year. From our mapping perspective these annual grasslands (set-aside fields or meadows) may still be considered as croplands since their dominant use over the time period is cropland. This areas also do not fulfil the criteria to be classified as permanent grasslands. Such differentiation of land uses is only possible using a robust modelling approach which considered more than a single year in the classification model. We have clarified this aim in lines 12, 41 and 58 (in the traced version of the manuscript) |
| About the features used for classification. I have seen all features used in this study is listed in Table 1, and I found features like ndvi_pc_05, ndvi_pc_25, so are these features  5th percentile of NDVI recorded from all three-year Sentinel-2 data? | Yes, eg. , the  "ndvi_pc_05" means 5th percentile of NDVI recorded from all three-year Sentinel-2 data. We mentioned this in the Tab.1 caption (see the new version of the manuscript). |
| Generally, we consider the time series characters for cropland/crop type mapping, for example, Low et. al (2015), Hao et. al (2018), I am not sure whether the features collected in this study have the potential to separate cropland, grassland and shrubland. | Thanks for this comment. We found that the time series parameters were crucial to distinguish between cropland, grassland, and shrubland. Using the observations from three different growing seasons, the parameters of the distribution of those features we collected were found as the most important in the modelling process. See section 3.1 of the results |
| Furthermore, Figure 4 showed the value range of indices of cropland, grassland, and shrubland, I suggest using some separability measurement methods, like JM distance to evaluate the separability. | We did not apply the separability measure as this are were only informative plots that helped us to justify the usability of the metrics. In the end the separability was defined through interaction of all metrics using the random forest model. Their importance is documented in the new Fig. 5. |
| And I am also concerned about whether the features of high separability are applicable in the entire study region. | The separability differed between the two regions (Jura and Plateau, and Alps) which demonstrates the different usefulness of the metrics between the two strata. The usefulness of each metric is then prioritized by the random forest model and documented in the new Fig. 5. |
| For parcel-level testing dataset validation, please show the location of the validation | Thanks for this comment. However, we described the distribution of the samples and |

| | |
|---|---|
| samples, and then show some validation examples to better clarify the validation samples. | validation procedure in the manuscript. We prefer to not include it as an illustration as the parcel-level testing was used as only one testing approach and we also think that the number of figures in the edited version of the manuscript is already relatively high. |
| Please show the confusion matrix of the validation samples, which could clearly indicate the misclassification samples | We included the confusion matrix into the manuscript. Please see Table 2. |
| Please compare your cropland/grassland map with some existing land cover map, such as FROM-GLC, GLC_FCS30 by wall-to-wall comparison. This can prove that your national outperformed the global products. | We compared our map with the three different datasets with the global coverage: PROBA-V LC, FROM_GLC and GLC_FCS30. The illustration is provided in Figure 8. |

| Reviewer #2 comments | Author's reply |
|---|---|
| The authors present an agricultural cropland, grassland and shrub map for Switzerland based on a random forest classification using optical Sentinel-2 metrics. The overall high impact of agriculture on biodiversity and landscape alteration (including the implied consequences on disaster risk and other domains) demand for a large scale understanding of land cover distribution and organisation in this field. The authors present Switzerland as a challenging case for a random forest based cropland, grassland and shrubland map. This seems plausible and makes the study an interesting case. | Thanks for this comment. We really appreciate your opinions and comments. |
| The authors describe the heterogeneous character of Switzerland, which makes it kind of unique among European countries of the temperate climatic zone, and you also point at strict landscape protection measures and a high demand for ecosystem services. Here, I would like to see a more detailed elaboration. To what degree is it heterogeneous? With regard to topography only? What is an example protective measure? Why is the | The description of the heterogeneous character of Switzerland and the agricultural protection measures has been expanded and is provided in lines 43-55 (in the traced version of the manuscript) along with associated references. |

| | |
|---|---|
| demand for ecosystem services high? As these environmental and regulatory conditions are a major reason for your innovation, this needs to become clearer. I would suggest to also include references for this | |
| Figure 1: How do the biogeographic regions differ? | The bioregions were defined based on statistical analysis of flora and fauna observational data. An explanation is now provided in lines 45-55. |
| Along with the previous comment: I would be happy to see more background information about the used methods. For example references that show the use of annual image metrics for land cover classification (e.g. Pflugmacher et al. 2019, doi: 10.1016/j.rse.2018.12.001). This can be short. | Some references have been added, see lines 120-127 (in the traced version of the manuscript) |
| - Line 50: Please use the full form of Google Earth Engine when using it for the first time. | Corrected, now line 61 (in the traced version of the manuscript) |
| I wonder whether the application of three-year Sentinel-2 metrics is applicable for agricultural mapping, especially with frequent crop rotation (which seems to be the case in Switzerland, according to your information). Particularly when grassland is part of the annual crop rotation. Isn't this exactly what you try to distinguish? I doubt if with data from three years, you can distinguish land cover that may change on an annual basis. I suggest using the same procedure with data from one year only, e.g. 2018, which was a rather cloud-poor year, and compare results. | The purpose of this study was to differentiate permanent grassland from the areas of different crops therefore we used the three-year time period 2017-2019. We have clarified this aim in lines 12, 41 and 58 (in the traced version of the manuscript). In our model, those areas that were covered by temporary grasslands were likely classified as the croplands. This is in line with our classification approach since the annual grasslands were a part of the crop rotation cycle and had different ecological properties than the temporal grasslands.

Using the 3-year time period for classification allowed us to develop a more robust classification model and thus increase the classification accuracy. While developing the model we also tried a single year classification model. A single year classification model was found with lower accuracies of the output model then by using the 3-year time period. The likely cause of the lower accuracies of the single year classification model, as noted before, is the inclusion of annual grassland which is a part of the crop rotation cycle.  We include this explanation into the text. See lines |

| | 58 – 60 (in the traced version of the manuscript) |
|---|---|
| Considering your indices: Tasseled cap metrics are more robust to mapping vegetation in areas affected by shadows than NDVI metrics. The relief map suggests that shadows could be a frequent challenge in your area. It would be interesting to know if your mapping results are weaker, for example, north of a mountain range compared to south of it. | This is indeed an interesting idea. We agreed that the tasselled cap could give better results in complex and rough terrains. On the other hand, we believe that using NDVI and the different metrics related mostly to the distribution of its values over the growing season clarify the properties of different classes more precisely then the Tasselled cap and are more intuitive. |
| I am not a huge fan of thresholding. Please explain if the asusmption that non-vegetated areas can be identified by a 95th NDIV perc. could lead to misinterpretation when agricultural plots are fallow for a year. | Given crop rotation practices in Switzerland, we expect that any land fallow at a given time point within the 3-year data period would also return to agricultural use within the 3-year period and will therefore have a different phenology to the non-vegetated area. Again, in order to retrieve statistics from a robust sample, we construct the statistics over a 3-year period of Sentinel-2 measurements which was more robust than considering only a single growing season, which would not consider crop rotation. |
| Following RC1, I would also be interested to see some example testing sites from the different testing datasets. | We added an example of comparison of different datasets according to the RC1 comment. Please see Fig. 8. |
| - Figure 4: Is this from your training plots? | Yes. We added this information in the caption of Fig. 4. |
| Please discuss your selection of metrics. In Fig. 4, it seems like cropland and grassland in the Jura and Plateau region could as well be mapped with ndvi_pc_05 only, while BLUE could as well be left out. What does this mean for a potential transfer of the models? | Thanks for this comment. We added some discussion on single-metrics separability (Lines 225-236 (in the traced version of the manuscript)). Regarding the other indices we noticed that the value range in the boxplots in the original Fig.4 were largely dependent on the outliers, where a larger range including outliers narrowed the boxplot size and thus resulted in the perception of no differences between different LC classes for particular indices. We have now updated Fig.4 |
| Please discuss what using elevation means for the transfer of your model. I think that elevation could be a very specific variable for Swiss cropland and grassland, not applicable to other regions. | Thanks for this comment. We added few lines on some discussion to lines 239-243 (in the traced version of the manuscript). |
| I underline the comment of RC1: Please show a confusion matrix (maybe instead of Fig. 5) with the exact classification results. This helps to understand where the errors occur. | We added the accuracies in the error ma matrix. See Table 2. |

| | |
|---|---|
| In the beginning, you say that a nation-wide map can respond better to the specific demands of local model parametereization compared to a continental map. In line with RC1, I would ask you to show a comparison of your map wiht large area continental or global maps. | We provided an illustrative comparison of the results with global map products, please see the Fig. 8 and lines 284-289 (in the traced version of the manuscript). |
| Please explain your choice for two separate models. Have you tried using one single model for the whole study area? I would be interested to see how this performs and where it is comparatively strong/weak. | Before we ran the modelling on the two strata selected and described in this paper, we run several checks to identify appropriate stratification of the areas as we found the national-wide model biased in particular regions of Switzerland. Finally, we used two strata as we found lower accuracies of modelling the high alpine environment with the national-wide model particularly on the areas mixed with the alpine shrubland. The areas of shrubland (used to mask our cropland – grassland map) often represented within the transition between the forest, alpine grassland and bare soils was much better modelled for the Alps stratum than the national-wide model. We have added some discussion of this point, lines 136-140. |